# Place-Based Strategies Addressing Neighborhood Environments to Improve Perinatal and Preterm Infant Outcomes

**DOI:** 10.3390/children10101646

**Published:** 2023-10-02

**Authors:** Timothy D. Nelin, Kristan A. Scott, Allan C. Just, Heather H. Burris

**Affiliations:** 1Division of Neonatology, Department of Pediatrics, Children’s Hospital of Philadelphia, Philadelphia, PA 19104, USA; scottk8@chop.edu (K.A.S.); burrish@chop.edu (H.H.B.); 2Department of Pediatrics, Perelman School of Medicine, University of Pennsylvania, Philadelphia, PA 19104, USA; 3Center of Excellence in Environmental Toxicology, Perelman School of Medicine, University of Pennsylvania, Philadelphia, PA 19104, USA; 4Leonard Davis Institute of Health Economics, Philadelphia, PA 19104, USA; 5Department of Epidemiology, School of Public Health, Brown University, Providence, RI 02912, USA; allan_just@brown.edu

**Keywords:** air pollution, greenspace, PM_2.5_, preterm birth, bronchopulmonary dysplasia, environment

## Abstract

Preterm birth (defined as birth <37 weeks of gestation) is a significant health concern globally, with lasting implications for individuals, families, and society. In the United States, high preterm birth rates among Black and low-income populations likely result from differences in environmental exposures. Structural racism and economic disadvantage have led to unequal distribution of polluting industrial sites and roadways across society as well as differential access to health-promoting resources which contribute to preterm birth risk. Once born, preterm infants remain at risk for numerous environmentally responsive adverse health outcomes that affect growth and development throughout childhood and adulthood. In this commentary, we describe associations of neighborhood environments with pregnancy and preterm infant health outcomes and propose strategies to address harmful exposures that affect families across the lifespan.

## 1. Introduction

In the United States, the preterm birth rate rose 4% in 2021 to 10.49%, after an overall decline from 2007 to 2020 [1], with persistent significant racial disparities. The preterm birth rate is 50% higher for Black compared to White individuals [1,2,3]. Black infants are also three times more likely to be born <28 weeks’ gestation, which represents the most vulnerable neonatal population for multiple medical morbidities and mortality [4]. Racism, not race, contributes to racial disparities in preterm birth rates in the United States through unequal exposures to environmental factors, such as air pollution, violence, crowded housing, low levels of greenspace, and discriminatory policies [2,5,6]. Many of the same environmental exposures that are associated with preterm birth are also associated with adverse health outcomes after birth, particularly among preterm infants. Understanding the relationship of the environment with perinatal health is a vital step in identifying modifiable factors and developing place-based strategies to reduce inequities in preterm birth risk and long-term adverse health outcomes of preterm infants after discharge from the neonatal intensive care unit (NICU).

Preterm infants are at risk for numerous adverse outcomes, including, but not limited to, neurodevelopmental impairment, bronchopulmonary dysplasia (BPD), long hospital lengths of stay, and increased healthcare utilization throughout childhood [7,8,9,10]. BPD represents the most common chronic morbidity of preterm birth and is a lifelong disease with extensive healthcare burden and cost for patients and families [11,12,13]. Significant racial disparities exist in early life outcomes of infants born very preterm; Black preterm infants have a higher mortality rate (12.9% vs. 11.7%) and higher readmission rate (23.2% vs. 18.9%) compared to White preterm infants [14]. It is important to note that racism, not race itself, contributes to the racial disparities present among preterm infants [15]. The environment, dynamically shaped over time through policies, migration, and external factors, is a compilation of multiple exposures that may modify the risks of adverse perinatal health outcomes. Examining environmental contexts is crucial to identify at-risk families, address underlying drivers of adverse outcomes, and develop place-based interventions to optimize perinatal health and equity.

Environmental exposures may be broken down into micro-environmental and macro-environmental factors [2]. Micro-environmental exposures are specific to the individual’s non-genetic exposures and may include diet, physical activity, smoking, and social relationships. Macro-environmental exposures are those that are shared by a community and over which individuals may have less control, such as ambient air pollution, extreme heat waves, neighborhood violence, and other community characteristics. Importantly, these exposures do not happen in isolation as the macro-environment may influence the micro-environment. For instance, an individual’s residence in a community that is a ‘food desert’ may affect dietary intake. We will focus on measured exposures of the macro-environment, specifically the physical environment including neighborhood characteristics and air pollution.

A growing body of literature demonstrates associations of neighborhood environments with adverse health outcomes [16,17]. Numerous indices have been developed to characterize neighborhood environmental characteristics that may affect health including, but not limited to, the Centers for Disease Control and Prevention (CDC) Social Vulnerability Index (SVI), the Area Deprivation Index (ADI) developed at the University of Wisconsin, the Child Opportunity Index, and the Community Material Deprivation Index developed at the University of Cincinnati [18,19,20,21]. Each of the metrics incorporates various factors such as household size and composition, income, poverty levels, education, and insurance status to describe neighborhood deprivation and vulnerability. Understanding the relationship of the neighborhood environment with pregnancy-related outcomes and preterm infant health after discharge from the NICU is central to the ability to develop interventions to improve perinatal health outcomes.

In this commentary, we will highlight known associations of the macro-environment with both preterm birth risk and preterm infant health outcomes and discuss potential interventions to improve health and address inequities.

## 2. Neighborhoods and Preterm Birth

### 2.1. Neighborhood Deprivation

Numerous studies demonstrate associations of neighborhood disadvantage with preterm birth. A 2016 systematic review and meta-analysis of cross-sectional studies investigating poverty, deprivation, racial residential segregation, and crime in the United States found a 27% higher risk of preterm birth among the most disadvantaged neighborhoods when compared to the least disadvantaged neighborhoods [22]. In a 2023 retrospective cohort study using the US Vital Statistics data, Salazar et al. reported an association of the Maternal Vulnerability Index (MVI)—comprised of 43 area-level indicators reflecting the physical, social, and health care environments of mothers and birth parents—with preterm birth [23]. The MVI ranged from 0 to 100 and residing in a county with the highest MVI level (80–100) was associated with 18% elevated adjusted odds of extreme preterm birth compared to the lowest MVI (0–20), even after adjusting for individual level confounding variables such as race and ethnicity, insurance, maternal education, and maternal health characteristics.

### 2.2. Residential Segregation

Longstanding structural residential segregation contributes to today’s racial disparities in neighborhood conditions and thus in perinatal health. A cohort study in New York state investigated the effects of redlining during the Great Depression—a racially discriminatory home loan practice—and demonstrated higher preterm birth rates in ZIP codes historically defined as “Hazardous” by the Home Owners’ Loan Corporation map [24]. In a study of 376 counties in the United States, Mehra et al. found a 6.9 increase in the odds of preterm birth of Black women compared to White women residing in racially isolated counties relative to non-racially isolated counties [25]. Similarly, a cross-sectional study of birthing parents in Chicago, Illinois, found an association of 12% higher adjusted-odds of preterm births among Black women residing in redlined neighborhoods relative to non-redlined neighborhoods [26]. These findings support a causal pathway connecting profound structural inequities to racial disparities in perinatal outcomes.

### 2.3. Air Pollution

A key environmental exposure that varies by neighborhood is air pollution. Air pollution is a heterogeneous mixture of harmful particles and gases that cause adverse health outcomes through localized airway irritation as well as systemic inflammation [27,28]. The Environmental Protection Agency (EPA) has identified and regulates six “criteria” air pollutants that are associated with poor human health outcomes: particulate matter, carbon monoxide, lead, nitrogen oxides, ozone, and sulfur dioxide [29].

Herein, we focus on fine particulate matter smaller than 2.5 µm in diameter (PM_2.5_) which is the product of vehicle emissions, power plants, burning of fuels, tobacco smoke, and cooking. PM_2.5_ exposure is not equally distributed across society. Air quality can vary drastically neighborhood to neighborhood, city to city, state to state, and country to country [30,31,32,33,34]. In the United States, communities of color are disproportionately exposed to PM_2.5_ [6].

A wealth of animal and human studies on small and large scales link air pollution exposure during pregnancy to adverse birth outcomes, including preterm birth. A 2012 systematic review and meta-analysis by Stieb et al. reported increased odds for preterm birth attributable to increased PM_2.5_ exposure, with predominant effects from third trimester exposures [35]. A more recent systematic review and meta-analysis from 2017 by Li et al. also demonstrated increased odds of preterm birth with interquartile increases in PM_2.5_ exposure throughout pregnancy [36]. Lastly, Bekkar et al. investigated increases in air pollution exposure and extreme heat exposure related to climate change in a 2020 systematic review and meta-analysis [37]. Among over 30 million live births, PM_2.5_ was significantly associated with increased risk of preterm birth. Importantly, the authors reported that the populations most affected included Black birthing people and birthing people with medical comorbidities such as asthma. Given the concurrent racial disparities in preterm birth and air pollution exposure, reducing PM_2.5_ levels represents a critical target to reduce preterm birth and improve equity.

## 3. Neighborhoods and Preterm Infant Health

### 3.1. Neighborhood Environment and Outcomes of Infants with BPD

While there is a paucity of studies focused on neighborhoods and outcomes of preterm infants overall, several studies have examined associations of neighborhoods and outcomes among infants with BPD. Banwell et al. performed a retrospective cohort study in the metropolitan Philadelphia and Baltimore regions among infants with BPD to investigate the association of ADI with respiratory health outcomes in these infants. The authors identified an association of ADI with readmission, emergency department visits, and activity limitations among infants with BPD [38]. Additionally, the authors found that infants with discharge addresses in the highest tertile of ADI were more likely to be Black, publicly insured, and born with lower gestational ages and birth weights [38]. This study emphasizes two important points: (1) families that live in more deprived neighborhoods are more likely to have preterm infants and (2) preterm infants are more likely to be discharged home to deprived neighborhoods. Furthermore, the strong associations of neighborhood deprivation with race and insurance status support that publicly insured and Black families likely have higher rates of preterm birth and that their preterm children have worse outcomes than their White counterparts partly due to differences in environmental exposures.

Others have also found that environments are associated with post-discharge outcomes of infants with BPD. In 2021, Deschamps et al. investigated neighborhood disadvantage and respiratory outcomes in the first year after NICU discharge in France. Among infants with BPD, infants living in disadvantaged areas had higher risk for hospital readmission [39]. After adjusting for perinatal and discharge-related clinical characteristics and season of birth, the authors found that respiratory-related readmissions were nearly 3-fold higher among infants living in disadvantaged areas [39]. Members of our team also investigated the association of SVI with medically attended respiratory illness in the first year after NICU discharge among 378 infants with BPD in the Philadelphia area [40]. After adjusting for gestational age, infant sex, birth year, BPD severity, and insurance status, higher SVI was associated with elevated odds of medically attended acute respiratory illness in the first year after NICU discharge. Furthermore, while adjustment for race and ethnicity attenuated this relationship, SVI mediated (potentially explained) 31% of the Black–White disparity in emergency department visits [40]. These studies suggest that neighborhoods matter for respiratory health outcomes among preterm infants with BPD.

### 3.2. Neighborhood Environment, Infant Mortality, and Development

Associations of the neighborhood environment with neonatal mortality, neurodevelopmental outcomes, and follow-up clinic attendance also exist [41,42,43]. Janevic et al. found that racial and economic neighborhood segregation was associated with neonatal mortality, specifically finding that infants residing in neighborhoods with the highest relative concentration of Black residents in New York City had higher neonatal mortality risk compared to infants in neighborhoods with the highest relative concentration of White residents [42]. Nwanne and colleagues found that infants from high-risk neighborhoods, defined as a maternal residential census block group with a score ≥75th percentile on a neighborhood risk index constructed using eight measures from the 2012–2016 American Community Survey (ACS), were more likely to have neurodevelopmental impairment, lower markers of cognitive function, and lower reading scores compared to infants from low-risk neighborhoods [43]. Similarly, Fraiman et al. quantified disparities in infant follow-up participation and attendance between Black infants and White infants, infants of non-English speaking compared to infants of English-speaking families, and infants with a home address in an area of very-low Child Opportunity Index compared to a home address in an area of very-high Child Opportunity Index [41]. Taken together, there are numerous associations of neighborhood disadvantage with adverse health outcomes among preterm infants, raising the question as to which neighborhood interventions could improve the health of preterm infants most substantially.

### 3.3. Air Pollution Exposure and Infant Outcomes

Air pollution not only increases the risk of preterm birth, but also adversely affects the health of infants after birth. Mechanistically, this may be due to the direct and indirect effects of air pollution exposure and its ability to cause local inflammation and damage in the lung, but also to translocate across the pulmonary vasculature and precipitate systemic inflammatory responses [27,44]. Given the role of climate change and human factors in the increase in occurrence of natural air polluting phenomena such as wildfires, the racial disparities inherent in air pollution exposure, and the fact that the majority of individuals in the world are exposed to higher levels of air pollution than deemed “safe” by the World Health Organization (WHO) and EPA, it is increasingly important to understand how these exposures contribute to preterm infant health outcomes [6,45,46,47].

Building on the well-established line of evidence linking PM_2.5_ exposure with adverse cardiovascular and respiratory complications in adults and children [48,49,50,51], several recent studies have investigated the association of PM_2.5_ exposure with respiratory outcomes in preterm infants [52,53,54,55,56]. Teyton et al. performed the largest retrospective cohort study, to date, of 1,983,700 term and preterm infants in California. The authors found an association of each 5 µg/m^3^ increase in PM_2.5_ exposure with increased odds of infection-related and respiratory-related emergency department visits among term and preterm infants. The stronger association was among preterm infants, suggesting that preterm infants may be particularly sensitive to worse air quality than their term counterparts [52].

These findings are consistent with less granular studies that have investigated proxies of air pollution exposure, namely residential proximity to a major roadway, environmental tobacco smoke exposure, and indoor combustion exposure and their relationship with respiratory health among preterm infants. Collaco et al. found that proximity to a major roadway and traffic-related air pollution exposure was associated with activity limitations and nighttime cough among infants with BPD [53]. In a separate study, Collaco et al. reported an association of higher hair nicotine levels among infants with BPD residing in households with caregiver reported smoking and an association of higher hair nicotine levels with increased hospitalizations and decreased activity levels [56]. Rice et al. demonstrated an association of exposure to combustible sources of indoor air pollution (i.e., gas or propane heat, gas or wood stove, gas or wood burning fireplace) with increased risk of hospitalization and chronic respiratory symptoms including activity limitation and cough among infants and children with BPD [54].

### 3.4. Air Pollution and Infant Neurodevelopment

Numerous studies suggest that prenatal exposure to air pollution is associated with adverse neurodevelopmental outcomes in preterm infants [57,58]. Similarly, evidence exists highlighting an association of postnatal air pollution exposure and adverse neurodevelopmental outcomes, including autism, in term infants and children [59,60]. The relationship between postnatal air pollution and preterm infant neurodevelopmental outcomes has not been fully explored but represents an important area for further study and a potential avenue to intervene to improve outcomes for this particularly vulnerable population.

## 4. Neighborhood Strategies to Improve Outcomes; a Focus on Greenspace and PM_2.5_

Given the clear associations of neighborhood deprivation and air pollution with perinatal and preterm infant health outcomes, it is of utmost importance to develop strategies to mitigate these exposures. Furthermore, given residential segregation and disparities in exposures and outcomes, addressing neighborhood health will be required to improve health equity. There are numerous opportunities to address these outcomes at an individual and policy level. Urban planning efforts, tax incentives for positive local businesses, and building and supporting community centers to enhance community engagement may all improve perinatal health.

### 4.1. Greenspace

Greenspace, which has been shown to confer multiple health benefits, is another neighborhood factor that can be shaped and influenced by public policy to improve health. A 2020 systematic review and meta-analysis found an association of increased residential greenspace with higher birth weight, lower odds of small for gestational age birth, and a non-significant association with lower odds of preterm birth [61]. A retrospective cohort study of nearly four million births in California found an association of increased greenspace with decreased risk of overall preterm birth, with the strongest association for very preterm birth (28–<32 weeks gestation) [62].

On an individual level, numerous studies have investigated greenspace and health outcomes in the perinatal and postpartum periods. Tiako et al. found that residential tree canopy coverage was associated with reduced stress among pregnant people in urban settings [63]. South et al. performed a pilot randomized trial to increase greenspace among postpartum individuals and found that among trial participants, those in the intervention arm had more and longer visits in nature and a non-significant trend towards lower post-partum depression scores [64]. Given the association of greenspace with reduced risk of preterm birth [65,66] and the reported relationships of increasing access to greenspace with improved maternal mental health outcomes, efforts to increase community greenspace represents a promising intervention. Greenspace may also serve to improve overall community health and reduce exposure to other factors—such as stress and crime—that have been associated with an increased risk of preterm birth [67,68,69].

Greenspace may also encourage physical activity [70]. There is evidence that specific elements of neighborhood health contribute to maternal health and risk of preterm birth. Kash et al. investigated neighborhood walkability as a risk factor for preterm birth phenotypes in Philadelphia and found that walkability—as defined by publicly available Walk Score ranking—was associated with decreased odds of medically indicated preterm birth (i.e., delivery due to maternal or neonatal complications) [71]. However, walkability is not evenly distributed throughout cities and may not represent the same set of positive influences in largely Black or White neighborhoods. Indeed, in this study, the association of walkability was only protective for White individuals, suggesting that walkability alone, without safety, is insufficient to reduce inequities; in Philadelphia, neighborhood violence levels are higher in neighborhoods with a higher proportion of Black residents [72].

### 4.2. Addressing Air Pollution

Given the relationships among PM_2.5_ exposure, preterm birth, and adverse health outcomes of preterm infants, identifying, investigating, and implementing interventions to reduce PM_2.5_ exposure is critical to improving health. Policy interventions may represent the most effective intervention in reducing PM_2.5_ exposure. Importantly, the United States EPA sets and reviews air quality standards under the Clean Air Act, which was passed in 1963, amended in 1990, and most recently amended in 2022 as part of the Inflation Reduction Act. From 2000 to 2022, the EPA has reported a 42% decrease in PM_2.5_ nationally [73]. Given the success in the Clean Air Act to decrease in PM2.5 nationally over the past 22 years, policies to improve air quality work. Continued advocacy for policies such as the Clean Air Act, emissions standards requirements, and renewable energy is an important opportunity to improve environmental health for birthing people and preterm infants [74]. Furthermore, thoughtful design of childcare facilities and their geographic relationship to major roadways and sources of air pollution may represent an additional opportunity to limit potentially harmful exposures to PM_2.5_.

Interestingly, community-level interventions, such as increasing greenspace and urban vegetation, may also serve to reduce air pollution in these settings. Greenspace is a unique intervention in that it provides green havens in otherwise urban environments that can be used for leisure and recreation. Trees and urban vegetation also play an independent role in removing air pollution from the environment [75,76,77,78]. Furthermore, studies demonstrate an association of lower overall and nighttime temperatures with increased tree canopy and urban vegetation [79,80]. These findings demonstrate the dual roles of greenspace to directly improve health while also decreasing levels of air pollution and extreme heat which can increase risks of preterm birth and adverse infant outcomes. Future research is needed to study the effects of greening interventions on preterm birth risk and preterm infant respiratory and developmental health to determine whether such interventions could improve perinatal health.

In addition to policies such as the Clean Air Act and greening interventions, at an individual family level, air purifiers represent an option that may be a feasible intervention to improve health outcomes. Air purifiers can reduce indoor air pollution exposure and have been shown to effectively reduce indoor PM_2.5_ concentrations [81,82]. Furthermore, the use of air purifiers has been shown to significantly improve allergic symptoms in children with allergic rhinitis [83]. A pilot study in Baltimore also demonstrated feasibility of randomizing women with children less than one year of age to receive air purifiers and secondhand smoke education, finding that the families receiving the air purifiers had significantly decreased PM_2.5_ levels and high participant satisfaction [84]. Similarly, interventions aimed at decreasing indoor air pollution sources, such as installation of electric or clean-burning stoves in the place of biomass or gas combustion as well as improved ventilation, have demonstrated improvements in respiratory health in children in developing countries and rural areas in the United States [85,86]. Interventions to reduce indoor air pollution through purification, ventilation, and decreased combustion may represent feasible interventions for families with preterm infants to optimize their respiratory and overall health.

## 5. Healthcare Interventions to Overcome Structural Inequities

Neighborhood improvements take time. As a healthcare community, we can intervene to overcome structural inequities and address some unmet needs and barriers to self-care and health care. These strategies may buffer some of the adverse neighborhood exposures by supporting families through stressors that are common in under-resourced communities.

Partnering with birthing people to improve access to and interaction with perinatal care providers may represent a strategy that can mitigate preterm birth risk in at-risk populations. Doulas are trained professionals who provide continuous social support to their patients before, during, and after the childbirth process [87]. Community doulas often come from the same communities as birthing people and thus can often provide information and resources that are accessible to families in addition to advocating for birthing people during healthcare visits [88]. Evidence suggests that doula care is linked with improvements in numerous perinatal outcomes [89,90]. Given potential disparities in access to doula care, two recent studies highlighted the results of implementing doula supports in populations of low-income women and women of color. Thomas et al. reported lower rates of preterm birth (6.3 vs. 12.4%, *p* < 0.001) among women receiving doula support in the New York City Department of Health and Mental Hygiene’s Healthy Start Brooklyn initiative compared to women not receiving doula support [91]. Kozhimannil et al. compared birth-related outcomes for Medicaid recipients who received doula care with Medicaid recipients receiving no doula care and found lower cesarean and preterm birth rates [92]. Improving access to doula care in at-risk populations may help bridge the gap between an individual and the healthcare environment, providing emotional, physical, and informational support, and represents an exciting strategy to improve birth outcomes for birthing people from neighborhoods with fewer resources.

There are other opportunities for partnership with families at the healthcare level to disrupt pathways by which infants—often preterm infants—from disadvantaged areas suffer worse health outcomes. Interventions to fulfill the unmet needs of caregivers represent an opportunity to improve health outcomes by enabling holistic and family-centered infant care. For example, partnering with families to provide transportation to follow-up clinics may improve attendance and participation in follow-up care. However, simply providing rides may be insufficient. A systematic review from 2020 showed mixed results of subsidizing transportation on healthcare outcomes but noted that transportation assistance is more likely to be effective when offered in tandem with other interventions aimed at reducing social and economic barriers to health [93].

Interventions such as the use of Community Health Workers (CHWs) and housing improvements have also been shown to be beneficial in reducing disparities in outcomes related to pediatric diseases such as asthma [94,95]. A similar approach to preterm infants with BPD may be an avenue to improve respiratory health outcomes. Integrating CHWs into the care of infants with BPD and systematically targeting the home environment of infants with BPD to decrease exposure to respiratory irritants could improve the respiratory health outcomes of infants with BPD and would take significant investment from both the community and the healthcare system.

On a policy level, interventions such as cash assistance, in the form of the expansion of pre-existing benefits such as Earned Income Tax Credits and Child Tax Credits, as well as unconditional cash transfers to caregivers of term infants have been associated with decreased food insufficiency, reduced child poverty, and improved markers of brain activity [96,97,98,99]. However, not all studies demonstrate benefit as Mergerison et al. recently found an association of increased odds of low birth weight with exposure to Child Tax Credit advance payments administered from July to December 2021 [100]. However, studying the impacts of the expanded Child Tax Credit and Earned Income Tax Credit for families with preterm infants is needed to understand their impacts on the outcomes of this vulnerable population after birth.

Similarly, the benefits of unconditional cash transfers have not been sufficiently investigated among preterm infants, though an initial study found that the provision of unconditional cash transfers resulted in higher parental visitation, more skin-to-skin time, and higher rates of breastfeeding during NICU admission [101]. These policy-level interventions are poised to reduce disparities in health outcomes of preterm infants and should be studied rigorously in the future.

## 6. Conclusions

Both pregnancy-related and preterm infant health outcomes are influenced by the neighborhood environment. Neighborhood deprivation, air pollution, greenspace, and other community factors are intertwined exposures that affect health. Environmental disadvantage confers a risk of preterm birth and, importantly, those same preterm infants are often discharged to a high-risk neighborhood environment. Addressing structural inequities in environmental exposures will be required to reduce perinatal and preterm infant health inequities. Only with the implementation of numerous state-sponsored and healthcare-initiated, place-based strategies at the societal and individual level will meaningful improvements in perinatal health and health equity be achievable.

## Data Availability

No new data were created or analyzed in this study. Data sharing is not applicable to this article.

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
