# Peer review of "Place-Based Strategies Addressing Neighborhood Environments to Improve Perinatal and Preterm Infant Outcomes"

_children, 2023, doi:10.3390/children10101646_

Round 1
Reviewer 1 Report
This is an interesting review on neighborhood environments on perinatal and preterm infants outcomes and subsequently place-based strategies that could be realized.
However, the title should clearly declare that this is a review an not a study based on original data.
Moreover, it is well demonstrated the role of neighborhood environments on perinatal and preterm outcomes although these elements can not be evaluated separately from educational level, employment and access to perinatal care. Nevertheless, these areas that could be implemented in strategies for improving outcomes. These areas should be considered in a review.
Author Response
Point 1: “However, the title should clearly declare that this is a review an not a study based on original data.”
Response: We appreciate this thoughtful comment and suggestion. To increase transparency, the article is now specified as a commentary.
Change to manuscript: The article is specified as a commentary, not based on original data.
Line: 1
Point 2: “Moreover, it is well demonstrated the role of neighborhood environments on perinatal and preterm outcomes although these elements can not be evaluated separately from educational level, employment and access to perinatal care. Nevertheless, these areas that could be implemented in strategies for improving outcomes. These areas should be considered in a review.”
Response: This is an excellent point as there are many factors within a neighborhood that likely contribute to perinatal and preterm infant outcomes. Given the context of existing data and the nature of the commentary, we chose to focus on aspects of the environment that are measurable and associated with outcomes. These measurable variables include indices of neighborhood vulnerability, socioeconomic status, healthcare access, child opportunity, segregation, greenspace, and air pollution. We agree that the areas of educational level, employment, and access to perinatal care can be implemented in strategies and should be studied going forward.
Change to Manuscript: None
Line: N/A
Reviewer 2 Report
General comments:
The manuscript lacks a coherent structure or argument and needs developing before publication. After having read the first few paragraphs of the introduction, I expected the focus to remain on structural racism and preterm birth, but the manuscript was much broader and more ambitious, yet did not offer a detailed literature review of any of the topics discussed. This was particularly true of the topic of unconditional cash transfers, which should have received more attention in the manuscript as a potentially effective solution. The evidence is convincing that preterm birth is associated with inequality, but actually fixing inequality through strategies which literally reduce it, such as unconditional cash transfers, should theoretically be the best strategy for addressing problems associated with inequality. The strategies suggested by the authors, such as creating neighbourhood parks and reducing exposure to air pollution by providing people with air purifiers (which cost the user money in electricity) do not address the main problem: inequality in wealth and routes to wealth in the first place.
Specific comments:
It would be useful to first define preterm birth, particularly because one of the first examples given refers to extremely preterm birth (<28 weeks of gestation completed), and when describing studies ensure that you include how the authors were defining preterm birth.
The authors stress that race is associated with preterm birth due to effects of racism rather than due to race itself. This is an important point, and there is an existing theoretical framework addressing why preterm birth may be so ubiquitously associated by experiencing adversity (e.g., see Williams, T.C.; Drake, A.J. Preterm birth in evolutionary context: A predictive adaptive response? Philos. Trans. R. Soc. B Biol. Sci. 2019, 374, 20180121). Using this framework would emphasise the point that preterm birth follows from adversity, not race.
The macro versus micro-environment dichotomy is confusing. Is it necessary? Neighbourhood violence is carried out by individuals, so it could be considered as micro-environment. The individual behaviours listed also have large macro-environmental contributing factors. Many (most) very economically deprived neighbourhoods are ‘healthy food deserts’ where fresh produce is unavailable and also unaffordable, making diet less of an individual decision than it is in wealthier areas.
Overall, the manuscript would benefit from reducing its scope and increasing its depth of analysis of a smaller subset of the literature.
Author Response
General Comments: “The evidence is convincing that preterm birth is associated with inequality, but actually fixing inequality through strategies which literally reduce it, such as unconditional cash transfers, should theoretically be the best strategy for addressing problems associated with inequality. The strategies suggested by the authors, such as creating neighbourhood parks and reducing exposure to air pollution by providing people with air purifiers (which cost the user money in electricity) do not address the main problem: inequality in wealth and routes to wealth in the first place.”
Response: Thank you for this thoughtful critique. We agree that fixing inequality through the redistribution of wealth, whether it be in the form of unconditional cash transfers, expansion of certain policies such as the Child Tax Credit or Earned Income Tax Credit, or the adoption of certain Western European social democratic policies regarding taxation and provision of public goods and services, represents the most encompassing strategy to reduce socioeconomic disparities in health outcomes. However, we chose to focus the bulk of our commentary on place-based strategies that have been studied rigorously and implemented on smaller levels. These are strategies that acknowledge the problems associated with inequality and try to work within the system to alter the environment and improve perinatal and preterm infant outcomes. We agree that policy-level interventions should be studied rigorously and represent a key gap in knowledge. However, due to a paucity of current original data, the scope of the commentary, and goal to highlight strategies that can be implemented at an individual level (as opposed to a policy level), we focused the bulk of the discussion on place-based strategies to address the structured environment.
Changes to Manuscript: None
Specific Comment 1: “It would be useful to first define preterm birth, particularly because one of the first examples given refers to extremely preterm birth (<28 weeks of gestation completed), and when describing studies ensure that you include how the authors were defining preterm birth.”
Response: Thank you for this comment and opportunity to improve our manuscript. We added a definition of preterm birth in the abstract to provide context to the reader.
Changes to Manuscript: We added a preterm birth definition in the abstract.
Line: 13
Specific Comment 2: “The authors stress that race is associated with preterm birth due to effects of racism rather than due to race itself. This is an important point, and there is an existing theoretical framework addressing why preterm birth may be so ubiquitously associated by experiencing adversity (e.g., see Williams, T.C.; Drake, A.J. Preterm birth in evolutionary context: A predictive adaptive response? Philos. Trans. R. Soc. B Biol. Sci. 2019, 374, 20180121). Using this framework would emphasise the point that preterm birth follows from adversity, not race.”
Response: Thank you for this important comment. We agree that preterm birth follows from adversity, not race.
Changes to Manuscript: We added wording in the introduction to highlight this point and setup a framework for reading the remainder of the manuscript.
Lines: 31-35
Specific Comment 3: “The macro versus micro-environment dichotomy is confusing. Is it necessary? Neighbourhood violence is carried out by individuals, so it could be considered as micro-environment. The individual behaviours listed also have large macro-environmental contributing factors. Many (most) very economically deprived neighbourhoods are ‘healthy food deserts’ where fresh produce is unavailable and also unaffordable, making diet less of an individual decision than it is in wealthier areas.”
Response: Thank you for this thoughtful comment highlighting an important consideration. This is an excellent point and it is important to note that environmental exposures do not happen in isolation and the macro-environment often impacts the micro-environment.
Changes to Manuscript: To address this comment, we added further context to micro- and macro-environmental exposures and highlight how the two are often intertwined. We add language to clarify that we will focus on the measured exposures of the macro-environment including neighborhood characteristics and air pollution.
Lines: 60-65
Round 2
Reviewer 1 Report
The revision is not in line with what was underlined in the previous review